

# A single Spanish version of maternal and paternal postnatal attachment scales: validation and conceptual analysis

Anna Riera-Martín[1], Antonio Oliver-Roig[2], Ana Martínez-Pampliega[1], Susana Cormenzana-Redondo[1], Violeta Clement-Carbonell[3] and Miguel Richart-Martínez[2]

[1] Department of Social and Developmental Psychology, University of Deusto, Bilbao, Spain
[2] Department of Nursing, University of Alicante, Alicante, Spain
[3] Department of Health Psychology, University of Alicante, Alicante, Spain

## ABSTRACT

**Background:** Postnatal bonding constitutes a major process during the postpartum period, and there is evidence that bonding difficulties have negative consequences for parents' mental health and the child's development. However, the conceptualization of postnatal bonding presents inconsistencies, as well as problems in having instruments that encompasses the father figure. The objective was to adapt the maternal postnatal attachment scale (MPAS) and the paternal postnatal attachment scale (PPAS) to Spanish, to evaluate its validity and reliability and to analyze the construct dimensionality of both questionnaires from a gender perspective.

**Methods:** Instrumental design. In 2016–2017, a sample of 571 mothers and 376 fathers, with children between 6 and 11 months of age, responded to the Spanish version of MPAS and PPAS, respectively. After a process of translation-back-translation of the instrument, we empirically analyzed the internal consistency (Cronbach alpha, composite reliability (CR)) construct and concurrent validity (with regard to postpartum depression and dyadic adjustment). Additionally, we studied the instrument's content validity, using the Delphi methodology; and the differential analysis in both samples (mothers and fathers), examining the invariance.

**Results:** A short version of 15 items was obtained, common for mothers and fathers. The results of the Delphi methodology showed a 100% inter-judge agreement, highlighting the absence of differences in the adequacy of the items as a function of the parents' gender. Confirmatory factor analysis showed a good fit of three original factors proposed by the authors. The global Cronbach alpha coefficients in the total sample were adequate (mothers, 0.70; fathers, 0.78); and Cronbach alpha of each dimension in the case of mothers was 0.50 (Quality of bonding), 0.55 (Absence of hostility), and 0.60 (Pleasure in interaction); in the case of fathers, it was respectively 0.54, 0.64, and 0.72. CR of each dimension were: quality of bonding, 0.74 in mothers and 0.80 in fathers; absence of hostility, 0.93 in mothers and 0.94 in fathers; pleasure in interaction, 0.83 in mothers and 0.90 in fathers. With regard to the analysis of group invariance, the results revealed empirical evidence of configural and metric invariance. Concurrent validity showed moderate negative correlations for postnatal depression (mothers, $r = -0.41$, $p < 0.001$; fathers, $r = -0.38$, $p < 0.001$), and positive correlations for dyadic adjustment (mothers, $r = 0.39$, $p < 0.001$; fathers, $r = 0.44$, $p < 0.001$).

Corresponding author
Antonio Oliver-Roig,
antonio.oliver@ua.es

**Discussion:** A new version of the instrument was generated, with good psychometric properties, adequate for use both with mothers and with fathers. This scale helps evaluate postnatal maternal and paternal bonding, allowing to study it from within the family system, a necessary step forward to advance perinatal mental health.

## INTRODUCTION

During the first years of life, newborns depend for their survival and development on their caregivers (*Bowlby, 1969*), who will provide the child's environmental context (*De Cock et al., 2016*). During this period, the parent–child relationship is crucial for the child's biological, cognitive, emotional, and social development (*Belsky & Fearon, 2002*; *Feldman, 2007*). One of the most important processes in the postpartum period is the postnatal bonding between the parents and the baby (*Brockington, 2004*, *2011*).

In the current state of scientific literature, the conceptualization of postnatal bonding presents inconsistencies, both linguistic and epistemological, across authors and disciplines, generating confusion and disagreement in practice (*Bicking Kinsey & Hupcey, 2013*).

Linguistically, inconsistencies in the use and meaning of postnatal bonding are due to confusing the term with attachment, bonding and attachment being used interchangeably. Nonetheless, different authors have specifically differentiated the terms (*Altaweli & Roberts, 2010*; *Kennell & McGrath, 2005*; *Taylor et al., 2005*). For example, *Taylor et al. (2005)* comment that bonding is used to describe how the mother feels toward her baby; while attachment includes the behavior of the baby toward the mother. Epistemological inconsistencies are related to the directionality of the process, the domains that comprise it (affective, behavioral, and biological) and its temporality (*Bicking Kinsey & Hupcey, 2013*).

Despite these inconsistencies, there is some consensus about identifying the emotional component of postnatal bonding in its different definitions (*Brockington, Fraser & Wilson, 2006*; *Condon & Corkindale, 1998*; *Condon, Corkindale & Boyce, 2008*; *Figueiredo et al., 2009*; *Taylor et al., 2005*; *Van Bussel, Spitz & Demyttenaere, 2010*), in addition to being consistently operationalized by measuring instruments (*Bicking Kinsey & Hupcey, 2013*). Therefore, according to the above, bonding is the term adopted in this study, meaning the unique emotional tie established between parents and their baby in early childhood.

Insights from studies on the negative consequences of postnatal bonding disorders both for the infant and for the parents highlight the importance of this concept. First, concerning the consequences for the infant, some studies relate low postnatal bonding quality with complications in the child's socioemotional development, increased parental perception of the infant's difficult temperament (*De Cock et al., 2016*;

*Mason, Briggs & Silver, 2011*), problems in the development of executive functions (*De Cock et al., 2017*), and externalizing behavior difficulties at 18 months of age (*Hairston et al., 2011*). Secondly, as to the consequences for parents, it was found that low-quality bonding predicts greater parental stress (*De Cock et al., 2017*; *Mason, Briggs & Silver, 2011*), higher anxiety levels, less perception of the partner's support (*De Cock et al., 2016*), a negative effect on parenting skills and feelings of parental adaptation (*Müller, 1994*; *Siddiqui & Hägglöf, 2000*), as well as poorer quality of parent–child interactions, for example, maternal response to the baby's signals (*Hornstein et al., 2006*).

Due to the importance of bonding and the consequences it can have on the infant and the parents, in recent years, specific instruments have been developed for professionals of nursing, psychology, pediatrics, and primary care to detect and prevent problems in early bonding. Easy-to-use instruments that do not need much time or require sophisticated infrastructure to evaluate qualitative information are required (*Van Bussel, Spitz & Demyttenaere, 2010*). In this sense, self-reports are the most useful and applied evaluation strategies, and the following three are the most extensively employed to evaluate early emotional bonding between the mother and the newborn: the maternal postnatal attachment scale (MPAS: *Condon & Corkindale, 1998*), the Postpartum Bonding Questionnaire (*Brockington et al., 2001*) and the mother–infant bonding scale (*Taylor et al., 2005*). These three instruments provide a reliable and valid indication of the early emotional bond between a mother and her newborn infant (*Van Bussel, Spitz & Demyttenaere, 2010*).

Although mother–infant bonding has received much attention over the last decade, the study of the father's role in the context of early relations is still little addressed in research, despite studies that have determined its great impact on the infant's development (*De Cock et al., 2016*, *2017*; *Ramchandani et al., 2013*; *Sarkadi et al., 2008*). For example, *Scism & Cobb (2017)* have shown that father–infant bonding in the postpartum period reduces cognitive delay, promotes weight gain in premature infants, and improves the rate of breastfeeding. In addition, incorporating fathers into the study of postnatal bonding, allows analyzing this concept from a family system perspective.

Only one of the above instruments, the MPAS, is based on a theoretical framework that encompasses the father figure and provides an instrument adapted to fathers—the paternal postnatal attachment scale (PPAS: *Condon, Corkindale & Boyce, 2008*)—thereby incorporating a gender perspective in the study of postnatal bonding. Additionally, it has adequate psychometric characteristics, is easy to apply, and has been widely used in different studies (*De Cock et al., 2016*, *2017*).

These two scales, the MPAS and the PPAS, draw on the conceptualization of bonding as an emotional state ("love") of the parents toward their baby. This emotional state leads to a series of needs and dispositions directed toward the baby, which will determine paternal and maternal behaviors (*Condon, 1993*; *Condon & Corkindale, 1998*; *Condon, Corkindale & Boyce, 2008*). These dispositions or needs, according to *Condon, Corkindale & Boyce (2008)*, differ in men and women. In women, they focus on the pleasure of proximity, tolerance, the need for gratification and protection, and the acquisition of knowledge

(*Condon & Corkindale, 1998*). In men, the bonding indicators are patience and tolerance, pleasure in interaction, affection and pride (*Condon, Corkindale & Boyce, 2008*).

A review of the literature on the MPAS and the PPAS has identified several works related to the adaptation of these scales to other countries. The MPAS has been adapted for use in Italy (*Scopesi et al., 2004*), Belgium (*Van Bussel, Spitz & Demyttenaere, 2010*), Portugal (*Nunes et al., 2014*), the United States (*Feldstein et al., 2004*), and Iran (*Ghadery-Sefat et al., 2016*). With regard to the PPAS, it has been adapted for use in Portugal (*Pires et al., 2014*) and Turkey (*Güleç & Kavlak, 2013*). Various studies report appropriate values of reliability and validity but not all the studies have analyzed their structural validity (*Van Bussel, Spitz & Demyttenaere, 2010*; *Feldstein et al., 2004*; *Ghadery-Sefat et al., 2016*). Among those who have done so, inconsistencies have been found, as the three factor structure of the original scales could not be replicated (*Scopesi et al., 2004*; *Nunes et al., 2014*; *Pires et al., 2014*). In this sense, in the Italian version of the MPAS, six factors were obtained (*Scopesi et al., 2004*) and in the Portuguese version of the two scales, two factors were obtained in each scale (*Nunes et al., 2014*; *Pires et al., 2014*). The version of the PPAS adapted in Turkey is the only one to have managed to replicate the original three-factor structure (*Güleç & Kavlak, 2013*). All studies mentioned above were performed on the general population. No studies have been found on clinical populations.

In short, contributing to progress in perinatal mental health requires validated instruments that allow the early detection of problems in mother–infant and father–infant bonding. Currently, there is no instrument validated in Spanish to assess postnatal bonding in both parents, making it difficult to study bonding within the family system. As previously mentioned, only the MPAS and the PPAS have both versions but, equally in this case, there is some discrepancy around the instruments' structure.

Therefore, the present study had a twofold objective: to study the psychometric properties of the Spanish version of the MPAS and the PPAS and, at the same time, to contribute to clarifying the dimensions of the postnatal bonding construct in mothers and fathers.

## MATERIALS AND METHODS

### Design and procedure

This instrumental study is part of a larger cohort study related to positive parenting (the PIPP Project). The process of studying MPAS and PPAS was developed in three phases. In the first phase, the scales were translated into Spanish. In the second phase, the data was collected. In the third phase, the psychometric properties of the two scales were tested.

### Phase 1: procedure of translation and adaptation

The process of adaptation of the Spanish version of the MPAS and the PPAS began through a process of translation-back-translation. Two bilingual translators made the Spanish translation, and two different bilingual translators made the back-translation into English. The translators worked independently. After completing this process,

a committee of experts assessed the two versions in English (original and back-translated), making sure that both questionnaires were understandable and equivalent to the original. Finally, a pilot study was conducted with six men and six women. The main problem reported by participants was the difficulty due to the heterogeneous number of items' response options, and they proposed that all the items should have the same number of response options. Taking into account these findings and the recommendations of *Condon (2015)*, in the final version we decided to homogenize the number of response options to five for all the items of both scales.

### Phase 2: data collection procedure

After recruiting the participants, we collected their contact details and requested them to complete a battery of forms. Clinical data about childbirth and early puerperium were obtained through the clinical history at postpartum discharge. At 6 months postpartum, a battery of online questionnaires was sent, with a link containing a unique code for each participant that allowed direct access to a self-reported form, which could be completed using a web browser.

### Phase 3: testing psychometric properties

The analysis of the distribution of items was performed by estimating for each score value the mean, standard deviation, skewness index, the correlation of each item with the scale ($r$), and the value of the Cronbach alpha coefficient if the item were eliminated ($\alpha$-item). In order to facilitate reading and interpreting the data, all the dimensions were coded in such a way that a higher score indicated a higher intensity of the measured construct. Internal consistency was measured through Cronbach's alpha coefficient, with a value of 0.70 or higher regarded as acceptable (*Wu et al., 2011*).

We applied structural covariance techniques to perform confirmatory factor analysis (CFA) of the instrument's structure. We calculated the fit of our data to the measuring models using the AMOS program and the maximum likelihood method (*Arbuckle, 2014*). To analyze the goodness of fit of the model, we used: the root mean square error of approximation (RMSEA) and its confidence interval, considering values between 0.05 and 0.08 as acceptable, and <0.05 as very good; the goodness of fit index (GFI), and the comparative fit index (CFI), considering values higher than 0.90 as adequate (*Hair et al., 2006*).

As we did not replicate the authors' original structure or the subsequent proposed validation studies, we adopted a different approach, based on a study of the instrument's content validity and a differential analysis of the two versions of the instrument. Firstly, we studied the relevance of each item of the postnatal bonding construct using the Delphi methodology (*Landeta, 1999*). For this purpose, we invited a total of six experts in the field to perform their assessment. They agreed on the existence of a single version, common to both parents. Secondly, we performed CFA of this resultant single version. Thirdly, internal consistency was measured using Cronbach's alpha coefficient for the global scale and for each subscale. Additionally, the composite reliability (CR) was calculated in order to obtain another indicator of the degree of internal consistency. This measure was calculated based on the standardized lambda coefficients ($\gamma$) and their respective measurement errors ($\delta$) resulting from the CFA, with

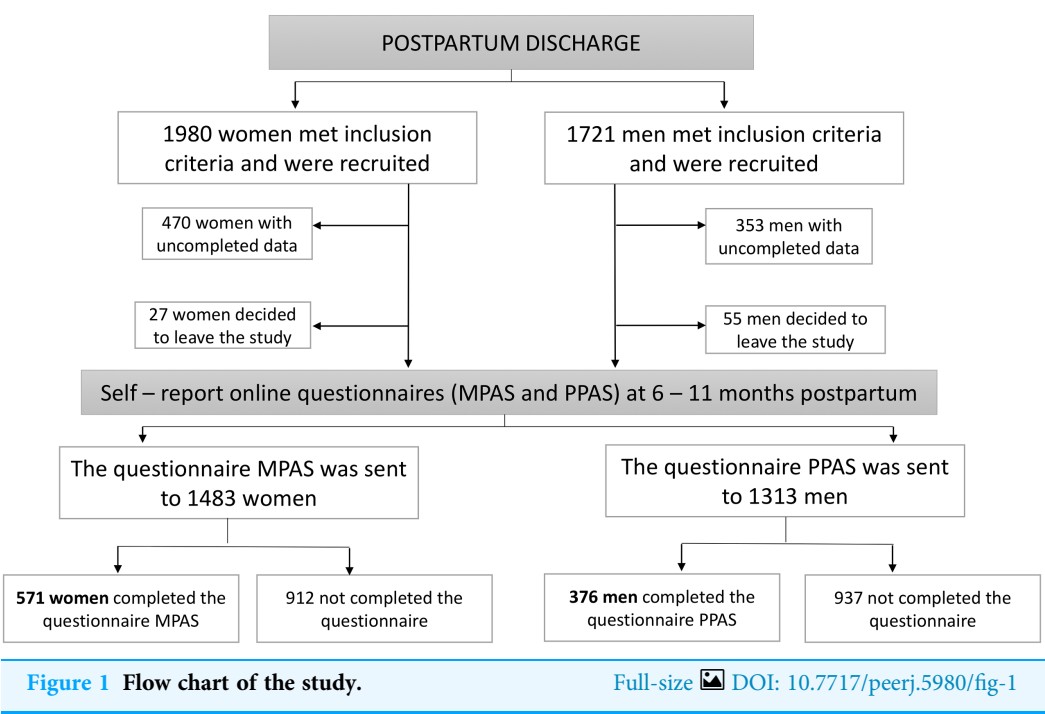

Figure 1 Flow chart of the study.     

the recommended values being equal to or greater than 0.70. Fourthly, we examined the invariance between the data from the application to mothers and fathers. For this purpose, we carried out the steps indicated by the model developed by *Little (1997)*, which establishes the need for the four types or models of invariance for a complex analysis: configural, metric, strong, and strict invariance. We performed multi-group comparison (for example, configural model vs. metric model). To interpret the results, we used the ΔCFI value. Finally, concurrent validity of the instrument was analyzed through Pearson correlations ($r$) with the variables postnatal depression and dyadic adjustment.

## Participants

Initially, the sample consisted of 1,980 women and 1,721 men recruited at postpartum discharge from the hospital. As inclusion criteria, all participants were parents of a full-term newborn baby, with low or medium risk pregnancies and births (according to the classification of obstetric risk of the Mother Care Program of local health council), who could speak and read Spanish without difficulty. Participants whose contact details were erroneous (353 men and 470 women) were excluded from the sample. Some participants decided to leave the study at 5 months postpartum (55 men and 27 women). Hence, 1,313 men and 1,483 women were included in the study, but only 571 women and 376 men completed the required questionnaires. Therefore, these data indicate a response rate of 38.5% for mothers, and 28.6% for fathers (see Fig. 1).

Mothers and fathers who participated in the study differed significantly from those who did not participate regarding the following characteristics: they were older, primiparous, Spanish nationality, higher educational level, and higher economic income. There were no

**Table 1 Characteristics of the study sample (n = 947).**

| | Mothers (n = 571) Mean ≠ SD(min–max) | Fathers (n = 376) Mean ≠ SD(min–max) |
|---|---|---|
| Age | 34.13 ≠ 4.26(20–47) | 35.98 ≠ 4.34(23–51) |
| | **n(%)** | **n(%)** |
| Marital status | | |
| Single | 89 (15.6) | 39 (10.4) |
| Married | 400 (70.1) | 253 (67.3) |
| Separated or divorced | 12 (2.1) | 7 (1.9) |
| Widow | 2 (0.5) | 0 (0) |
| Missing Values | 67 (11.7) | 77 (20.5) |
| Coexistence with the partner | | |
| Yes | 485 (84.9) | 292 (77.7) |
| Some days of the month | 9 (1.6) | 7 (1.9) |
| No | 9 (1.6) | 0 (0) |
| Missing Values | 68 (11.9) | 77 (20.5) |
| Education | | |
| Elementary school or lower | 16 (3.8) | 22 (5.9) |
| High school | 96 (16.8) | 85 (22.6) |
| High school senior or Vocational training | 109 (19.1) | 60 (16.0) |
| University | 279 (48.8) | 131 (34.9) |
| Missing values | 65 (11.4) | 78 (20.7) |
| Income level | | |
| <6,000 | 30 (5.3) | 14 (3.7) |
| 6,000–8,999 | 26 (4.6) | 8 (2.1) |
| 9,000–11,999 | 51 (8.9) | 15 (4.0) |
| 12,000–17,999 | 108 (18.9) | 58 (15.4) |
| 18,000–29,999 | 142 (24.9) | 100 (26.6) |
| 30,000–44,999 | 81 (14.2) | 61 (16.2) |
| 45,000–60,000 | 33 (5.8) | 27 (7.2) |
| More than 60,000 | 18 (3.2) | 12 (3.2) |
| Missing values | 82 (14.4) | 81 (21.5) |
| Nationality | | |
| Spanish | 524 (91.8) | 292 (77.7) |
| Other | 25 (4.4) | 14 (3.7) |
| Missing values | 22 (3.9) | 70 (18.6) |

**Note:**
SD, standard deviation.

significant differences in their marital status, nor according to whether they were living together with their partner.

Regarding the participants' characteristics, the women's mean age was 34.13 years, and the men's was 35.98 years. Most of the participants were of Spanish nationality, married and living with their partner, had university studies, and their income was between 12,000 and 30,000 euros a year (see Table 1).

## Instruments

### Postnatal bonding

The MPAS (*Condon & Corkindale, 1998*), and PPAS (*Condon, Corkindale & Boyce, 2008*) were used to evaluate postnatal parental bonding.

The MPAS is divided into three factors (*Condon & Corkindale, 1998*; *Condon, 2015*). The first factor is the quality of bonding (formerly called quality of attachment) (nine items), which consists of confidence and satisfaction in interaction with the infant (e.g., *"When I am with the baby and other people are present, I feel proud of the baby"*). The second factor is the absence of hostility (five items), defined as the absence of hostile feelings or anger toward the infant (e.g., *"When I am caring for the baby, I get feelings of annoyance or irritation"*). The third factor is pleasure in interaction (five items), defined as the desire for physical closeness and joy in interaction with the baby (e.g., *"When I am not with the baby, I find myself thinking about the baby"*). The psychometric properties of the original version of the scale have shown adequate internal consistency of the global scale (Cronbach alpha of 0.78) (*Condon & Corkindale, 1998*). However, the structure of the factors has not yet been adequately established (*Condon & Corkindale, 1998*). Concerning concurrent validity, only the global score was considered, evaluated by means of the variables depression and anxiety, among others. The results obtained were consistent with the bonding literature (*Condon & Corkindale, 1998*), that is, the higher scores in postnatal bonding, the lower scores in depression and anxiety.

The PPAS is divided into three factors (*Condon, Corkindale & Boyce, 2008*). The first factor is patience and tolerance (eight items), defined as the absence of irritability and other negative affect toward the baby, such as the lack of resentment about the impact of paternity (e.g., *"When I'm looking after my baby, I feel sad, frustrated or irritated"*). The second factor is pleasure in interaction (seven items), consisting of feelings of pleasure, satisfaction, and competence in real interactions with the baby (e.g., *"When I am with my baby, I feel . . . "*). The third factor is affection and pride (four items), representing more stable and lasting feelings and cognitions toward the baby, including a sense of ownership (my baby), a sense of pride, and feelings of affection toward the baby (e.g., *"In the last 3 months, I felt I have had no time for myself or to do things that I'm interested in"*). The results indicated adequate values of internal consistency in the global scale (0.81) and the different subscales (0.75, 0.71, and 0.71) (*Condon, Corkindale & Boyce, 2008*). Regarding concurrent validity, the results were as expected according to the theory (*Condon, Corkindale & Boyce, 2008*).

Both scales have 19 items, with two to five response options. However, some versions of the instrument have five response options (*Scopesi et al., 2004*), ranging from 1 (*low bonding*) to 5 (*high bonding*), in order to ensure equal weighting of the items. In this study, we decided to use this response scale, as mentioned in the section on translation and adaptation procedures.

### Postnatal depression

Edinburgh postnatal depression scale (EPDS: *Cox, Holden & Sagovsky, 1987*). In this study, we used the adapted version of *Garcia-Esteve et al. (2003)*, which maintains the

structure of the original scale. This is a self-administered 10-item scale designed to evaluate the presence of postpartum depression (e.g., *"I could laugh and see the funny side of things"*). Each item is rated on a 4-point scale (0–3), with a total score ranging from 0 to 30, where higher scores indicate greater presence of symptoms of postpartum depression. In this version, the identified cut-off point was 10–11 for the presence of postpartum depression (*Garcia-Esteve et al., 2003*). *Maroto, García & Fernández (2005)* presented the psychometric characteristics of the Spanish version, and determined high internal consistency (Cronbach alpha of 0.79) and a two-factor structure (sadness and anxiety).

### Dyadic adjustment

Dyadic adjustment scale (DAS: *Spanier, 1976*). This variable was studied through the Spanish version of *Santos-Iglesias, Vallejo-Medina & Sierra (2009)*. This version is a self-administered 13-item scale designed to assess the quality of the couple's relationship (e.g., *"How often do you and your partner argue?"*). Each item has five or six response options. Its structure is made up of three factors: satisfaction, consensus, and cohesion. The psychometric properties of the scale have shown an overall internal consistency of 0.83, and 0.73, 0.70, and 0.63 for the subscales of consensus, satisfaction, and cohesion, respectively, as well as adequate validity (*Santos-Iglesias, Vallejo-Medina & Sierra, 2009*).

## Ethical considerations

The study was approved by the Ethical Committee of Clinical Research of the General Direction of Public Health and Higher Center of Research in Public Health (CEIC-DSGSP/CSISP), attached to the Health Council of the Valencian Community. The participants were informed of the study and gave informed consent by signing a written document. Only the members of the research team had access to personal data, which were replaced in the forms and databases by an alphanumeric code for each participant, in order to guarantee confidentiality.

## RESULTS

### Reliability and validity of the MPAS and the PPAS

Analysis of the distribution of the items of the MPAS (Table 2) and PPAS (Table 3) showed a high and negative skewness, reflecting a grouping response tendency toward the highest scores. In this sense, high scores are indicators of higher postnatal bonding. In the case of the mothers' version, we observed an exception, with Item 17 showing positive skewness. In both instruments, items with a kurtosis index equal to or greater than 2 indicated a high homogeneity in the responses (*George & Mallery, 2010*).

The correlation of the items for both instruments was low-moderate (MPAS: between 0.215 and 0.487, and PPAS between 0.107 and 0.642). The MPAS obtained internal consistency, assessed by Cronbach's alpha, of 0.75 on the global scale; and of 0.63, 0.56, and 0.61 in the subscales of quality of bonding, absence of hostility, and pleasure in interaction, respectively. The PPAS obtained internal consistency of 0.83 for the global scale, and of 0.78, 0.62, and 0.58 for the subscales of patience and tolerance, pleasure in interaction, and affection and pride, respectively. The analyses showed no improvement of these reliability indices upon deleting any of the items from the subscales.

**Table 2 Item distribution and descriptive characteristics of MPAS.**

| | 1 | 2 | 3 | 4 | 5 | M | SD | S | Ku | $r°$ | α-item° | $r$ | α-item |
|---|---|---|---|---|---|---|---|---|---|---|---|---|---|
| MPAS03 | 0 | 0.2 | 0 | 4.0 | 95.8 | 4.95 | 0.232 | −6.42 | 55.1 | 0.282 | 0.619 | 0.289 | 0.747 |
| MPAS04 | 1.2 | 1.7 | 5.9 | 16.0 | 75.1 | 4.61 | 0.778 | −2.43 | 6.26 | 0.296 | 0.609 | 0.284 | 0.744 |
| MPAS05 | 0.3 | 1.6 | 4.9 | 32.1 | 61.1 | 4.52 | 0.696 | −1.65 | 3.41 | 0.420 | 0.570 | 0.382 | 0.736 |
| MPAS06 | 0.2 | 0.5 | 9.4 | 41.5 | 48.4 | 4.37 | 0.690 | −0.872 | 0.611 | 0.407 | 0.574 | 0.480 | 0.728 |
| MPAS07 | 0.2 | 0.2 | 0.7 | 11.8 | 87.1 | 4.86 | 0.411 | −3.70 | 19.8 | 0.291 | 0.609 | 0.298 | 0.744 |
| MPAS10 | 0.2 | 0 | 0.9 | 18.8 | 80.1 | 4.79 | 0.454 | −2.46 | 9.41 | 0.433 | 0.581 | 0.508 | 0.732 |
| MPAS14 | 0 | 0 | 1.7 | 14.8 | 83.4 | 4.82 | 0.430 | −2.28 | 4.61 | 0.215 | 0.621 | 0.246 | 0.746 |
| MPAS18 | 0 | 0.3 | 1.9 | 44.8 | 53.0 | 4.50 | 0.5557 | −0.654 | 0.104 | 0.346 | 0.593 | 0.295 | 0.743 |
| MPAS19 | 1.6 | 3.3 | 13.9 | 34.0 | 47.2 | 4.22 | 0.916 | −1.20 | 1.26 | 0.273 | 0.629 | 0.299 | 0.744 |
| MPAS01 | 0 | 0.7 | 8.4 | 42.0 | 49.0 | 4.39 | 0.669 | −0.791 | 0.103 | 0.422 | 0.464 | 0.565 | 0.721 |
| MPAS02 | 0 | 0 | 2.4 | 13.8 | 83.8 | 4.81 | 0.448 | −2.93 | 5.15 | 0.231 | 0.555 | 0.276 | 0.744 |
| MPAS15 | 0 | 0.7 | 6.4 | 18.6 | 74.2 | 4.66 | 0.628 | −1.85 | 2.77 | 0.291 | 0.525 | 0.434 | 0.732 |
| MPAS16 | 15.9 | 20.4 | 34.8 | 23.7 | 5.2 | 2.82 | 1.12 | −0.098 | −0.78 | 0.471 | 0.398 | 0.364 | 0.740 |
| MPAS17 | 47.0 | 29.6 | 12.5 | 5.9 | 4.9 | 1.92 | 1.13 | 1.23 | 0.769 | 0.296 | 0.544 | 0.204 | 0.760 |
| MPAS08 | 0 | 0 | 2.6 | 28.9 | 68.5 | 4.66 | 0.527 | −1.20 | 0.418 | 0.289 | 0.596 | 0.475 | 0.732 |
| MPAS09 | 2.3 | 11.7 | 26.8 | 31.2 | 28.0 | 3.71 | 1.07 | −0.434 | −0.62 | 0.420 | 0.581 | 0.196 | 0.759 |
| MPAS11 | 0 | 3.3 | 0 | 16.4 | 80.3 | 4.74 | 0.627 | −2.97 | 9.49 | 0.421 | 0.535 | 0.246 | 0.746 |
| MPAS12 | 0.2 | 0.5 | 6.3 | 39.0 | 54.0 | 4.46 | 0.653 | −1.08 | 1.38 | 0.487 | 0.499 | 0.464 | 0.730 |
| MPAS13 | 0 | 0.3 | 0.5 | 11.7 | 87.5 | 4.86 | 0.388 | −3.24 | 13.2 | 0.376 | 0.579 | 0.392 | 0.740 |

**Notes:**

α = 0.75; KMO = 0.833, Bartlett's sphericity test 1990, 44, $p < 0.001$.

1–5, number of response options; M, mean; SD, standard deviation; S, Skewness index; Ku, Kurtosis index; $r°$, item-subscale correlation; α-item°, reliability index of the subscale if the item is removed; $r$, item-global scale correlation; α-item, reliability index of the global scale if the item is removed.

The KMO index and Bartlett's sphericity test indicated that the data matrix was factorizable for both instruments, so we proceeded to perform CFA.

First, we verified the three-factor structure of the original model of the original version of the MPAS (*Condon & Corkindale, 1998*) and PPAS (*Condon, Corkindale & Boyce, 2008*). The results did not show adequate goodness of fit in either instrument. Although RMSEA and GFI indices had acceptable values, the CFI index did not reach the recommended cut-off value.

Secondly, we verified the two-factor structure factor of the Portuguese version of the MPAS (*Nunes et al., 2014*) and the PPAS (*Pires et al., 2014*), made up of 14 and 16 items, respectively. As in the former case, the CFI index did not reach the necessary cut-off point for the goodness of fit of the model to be considered adequate.

Lastly, a new analysis was performed, only in the case of the MPAS, to confirm the six-factor structure of the Italian version (*Scopesi et al., 2004*). However, again, the required values for the entire set of indices measured were not achieved (see Table 4).

## Reliability and validity of a single version for mothers and fathers

Due to the inadequacy of the prior confirmatory analyses, a panel of experts met to determine, on the one hand, the degree of relevance of the items of the two scales; on the other hand, the existence of differences between maternal and paternal bonding.

**Table 3 Item distribution and descriptive characteristics of PPAS.**

| | 1 | 2 | 3 | 4 | 5 | *M* | SD | S | Ku | *r*° | α-item° | *r* | α-item |
|---|---|---|---|---|---|---|---|---|---|---|---|---|---|
| PPAS01 | 0.3 | 1.3 | 21.1 | 43.5 | 33.8 | 4.09 | 0.786 | −0.460 | −0.29 | 0.642 | 0.733 | 0.564 | 0.817 |
| PPAS02 | 0 | 0.3 | 4.2 | 19.0 | 76.5 | 4.72 | 0.551 | −1.93 | 3.24 | 0.356 | 0.778 | 0.340 | 0.828 |
| PPAS06 | 0.8 | 1.6 | 11.9 | 36.7 | 49.1 | 4.32 | 0.803 | −1.19 | 1.58 | 0.631 | 0.734 | 0.629 | 0.813 |
| PPAS11 | 0 | 0.3 | 1.3 | 31.1 | 67.3 | 4.65 | 0.519 | −1.22 | 1.15 | 0.552 | 0.758 | 0.586 | 0.820 |
| PPAS13 | 0.5 | 4.2 | 18.2 | 52.2 | 24.8 | 3.97 | 0.804 | −0.674 | 0.580 | 0.457 | 0.764 | 0.621 | 0.814 |
| PPAS17 | 0 | 1.3 | 8.2 | 26.9 | 63.6 | 4.53 | 0.617 | −1.28 | 0.870 | 0.505 | 0.757 | 0.483 | 0.822 |
| PPAS18 | 8.4 | 20.6 | 35.6 | 27.4 | 7.9 | 3.06 | 1.07 | −0.143 | −0.56 | 0.462 | 0.772 | 0.382 | 0.829 |
| PPAS19 | 1.6 | 4.2 | 17.9 | 37.5 | 38.8 | 4.01 | 0.935 | −0.933 | 0.603 | 0.416 | 0.775 | 0.388 | 0.827 |
| PPAS04 | 0 | 0.8 | 10.3 | 67.0 | 21.9 | 4.10 | 0.587 | −0.257 | 0.827 | 0.266 | 0.601 | 0.345 | 0.828 |
| PPAS05 | 2.6 | 3.2 | 9.2 | 32.7 | 52.2 | 4.29 | 0.948 | −1.58 | 1.57 | 0.107 | 0.660 | 0.154 | 0.841 |
| PPAS08 | 0.3 | 1.1 | 6.1 | 40.1 | 52.5 | 4.44 | 0.681 | −1.21 | 2.02 | 0.413 | 0.561 | 0.494 | 0.821 |
| PPAS09 | 0 | 2.6 | 25.9 | 47.2 | 24.3 | 3.93 | 0.777 | −0.221 | −0.57 | 0.382 | 0.565 | 0.359 | 0.828 |
| PPAS10 | 7.7 | 17.4 | 39.1 | 21.4 | 14.5 | 3.18 | 1.12 | −0.055 | −0.56 | 0.324 | 0.592 | 0.409 | 0.828 |
| PPAS12 | 0 | 7.4 | 32.5 | 39.8 | 20.3 | 3.73 | 0.868 | −0.134 | −0.71 | 0.483 | 0.526 | 0.493 | 0.821 |
| PPAS15 | 0.3 | 2.6 | 17.7 | 44.3 | 35.1 | 4.11 | 0.804 | −0.639 | 0.061 | 0.444 | 0.544 | 0.475 | 0.822 |
| PPAS03 | 0 | 0 | 0.3 | 9.0 | 90.8 | 4.91 | 0.302 | −3.06 | 8.54 | 0.397 | 0.528 | 0.404 | 0.829 |
| PPAS07 | 0.3 | 0 | 1.1 | 18.2 | 80.5 | 4.79 | 0.471 | −2.75 | 11.9 | 0.371 | 0.507 | 0.325 | 0.829 |
| PPAS14 | 0.3 | 0.3 | 2.6 | 26.6 | 70.2 | 4.66 | 0.567 | −1.91 | 5.37 | 0.369 | 0.511 | 0.428 | 0.825 |
| PPAS16 | 0 | 0.3 | 6.1 | 28.5 | 65.2 | 4.59 | 0.617 | −1.28 | 0.870 | 0.398 | 0.511 | 0.492 | 0.828 |

**Notes:**
α = 0.83; KMO = 0.882, Bartlett's test of sphericity 1816, 56, *p* < 0.001.
1–5, number of response options; M, mean; SD, standard deviation; S, Skewness index; Ku, Kurtosis index; *r*°, item-subscale correlation; α-item°, reliability index of the subscale if the item is removed; *r*, item-global scale correlation; α-item, reliability index of the global scale if the item is removed.

**Table 4 Confirmatory factor analyses summary for the MPAS and PPAS.**

| | Model | χ2 | d*f* | *p* | GFI | CFI | RMSEA | RMSEA IC-90% |
|---|---|---|---|---|---|---|---|---|
| MPAS | 3F | 447.893 | 146 | 0.000 | 0.92 | 0.84 | 0.060 | (0.054–0.066) |
| | 6F | 457.000 | 137 | 0.000 | 0.92 | 0.83 | 0.064 | (0.057–0.070) |
| | 2F-14 | 315.035 | 76 | 0.000 | 0.92 | 0.79 | 0.074 | (0.066–0.083) |
| | 3F-COM | 196.005 | 83 | 0.000 | 0.96 | 0.91 | 0.049 | (0.040–0.058) |
| PPAS | 3F | 390.265 | 146 | 0.000 | 0.90 | 0.86 | 0.067 | (0.059–0.075) |
| | 2F-16 | 282.954 | 103 | 0.003 | 0.91 | 0.87 | 0.068 | (0.059–0.078) |
| | 3F-COM | 197.754 | 83 | 0.000 | 0.93 | 0.90 | 0.060 | (0.049–0.071) |

**Note:**
$\chi^2$, chi-square; d*f*, degrees of freedom, *p*, probability; GFI, Goodness of fit index; CFI, comparative fit index; RMSEA, root mean square error of approximation; RMSEA (IC-90%), RMSEA confidence interval; 3F, original version of three factors; 6F, Italian version; 2F-14, Portugal version; 2F-16, Portugal version; 3F-COM, single version for fathers and mothers with 15 items.

Six experts participated in the study, all of them researchers in the concept of postnatal bonding, four of whom also had a clinical profile, with a long history of working with infants and families. The results showed a 100% inter-judge agreement, highlighting the absence of differences in the adequacy of the items as a function of the parents' gender. Therefore, we continued the analysis, using a common version for mothers and fathers, called postnatal bonding scale (PBS), which was made up of 15 items, after deleting

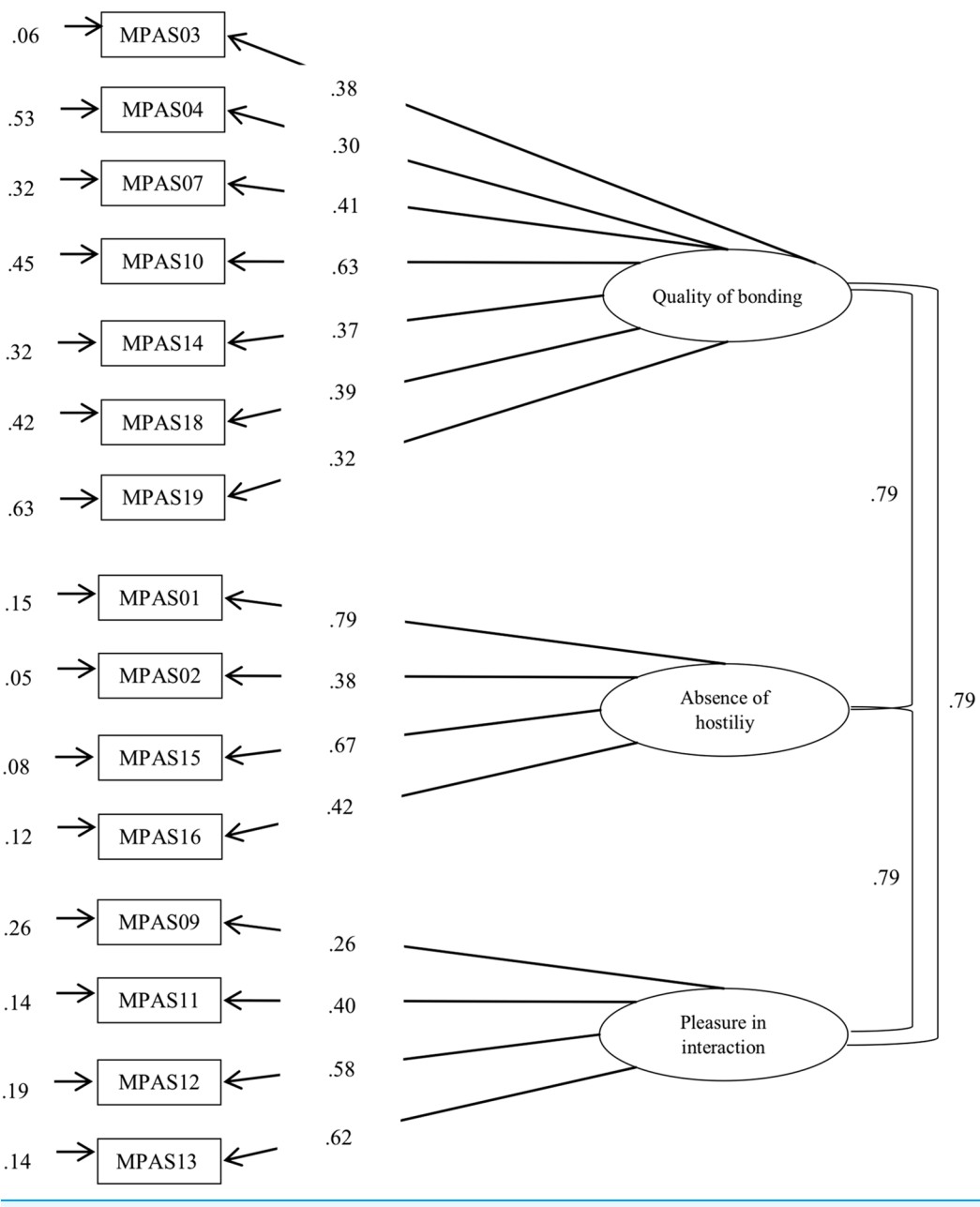

**Figure 2 Factor analysis of PBS by including MPAS items.**

the unshared items (items 5, 6, 8, and 17 of the mother's version, and items 6, 8, 9, and 15 of the father's version).

Firstly, a confirmatory analysis was performed of the new version, using as reference the original three-factor version of the MPAS, the first instrument developed by the authors (see Fig. 2, for mothers; Fig. 3, for fathers). The results showed an adequate fit (Table 4), as their indexes reached the recommended values (in mothers: GFI = 0.96, CFI = 0.91, RMSEA = 0.049; in fathers: GFI = 0.93, CFI = 0.90, RMSEA = 0.060).

The data obtained showed how the correlation coefficients between the three subscales took high and significant values, showing a high interrelation between the components of

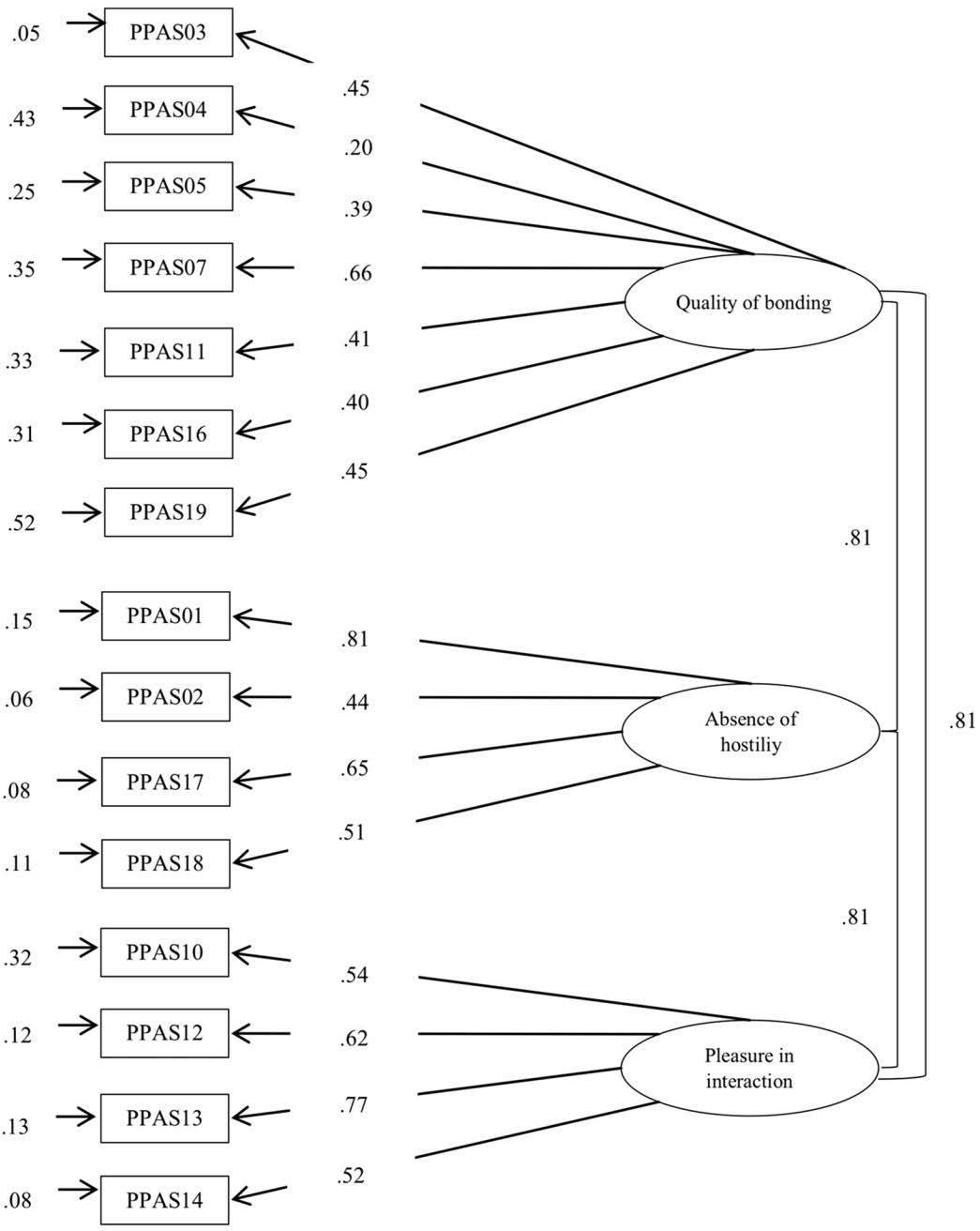

**Figure 3 Factor analysis of PBS by including PPAS items.**

the instrument: quality of bonding and absence of hostility (Mothers: $r = 0.81$, $p < 0.001$; Fathers: $r = 0.84$, $p < 0.001$) quality of bonding and pleasure in interaction (Mothers: $r = 0.83$, $p < 0.001$; Fathers: $r = 0.73$, $p < 0.001$), and absence of hostility and pleasure in interaction (Mothers: $r = 0.66$, $p < 0.001$; Fathers: $r = 0.53$, $p < 0.001$). All the standardized coefficients were statistically significant and obtained a value near or greater than 0.40, with the exception of Items 4 ($\lambda = 0.30$), 9 ($\lambda = 0.26$), and 19 ($\lambda = 0.32$) in the case of mothers; and of Item 4 ($\lambda = 0.20$) in the case of fathers. Lastly, the internal consistency

**Table 5  Fit indices for invariance tests.**

|  | $\chi^2$ | d$f$ | $p$ | CFI | $\Delta$CFI | RMSEA | RMSEA 90% CI |
|---|---|---|---|---|---|---|---|
| **Mothers–Fathers** | | | | | | | |
| Configural | 376.634 | 166 | 0.000 | 0.905 | – | 0.037 | (0.032–0.041) |
| Metric | 395.052 | 176 | 0.000 | 0.901 | 0.004 | 0.036 | (0.031–0.041) |
| Scalar | 542.416 | 185 | 0.000 | 0.838 | 0.063 | 0.045 | (0.041–0.049) |

**Note:**
$\chi^2$, chi-square; d$f$, degrees of freedom, $p$, probability; CFI, comparative fit index; $\Delta$CFI, change of the comparative fit index; RMSEA, root mean square error of approximation; RMSEA (IC-90%), RMSEA confidence interval.

index, analyzed through Cronbach's alpha coefficient, was 0.70 for mothers and 0.78 for fathers. Regarding the subscales, the strategies used were the Cronbach's alpha coefficient and the CR. The results obtained were as follows: in the case of the mothers $\alpha = 0.50$ and CR = 0.74 for quality of bonding; $\alpha = 0.55$ and CR = 0.93 for absence of hostility, and $\alpha = 0.60$ and CR = 0.83 for pleasure in interaction. In the case of fathers, results were respectively: $\alpha = 0.54$ and CR = 0.80, $\alpha = 0.64$ and CR = 0.94 finally $\alpha = 0.72$ and CR = 0.90. With regard to the analysis of group invariance (see Table 5), the results revealed empirical evidence of configural (Model 1) and metric invariance (Model 2) when comparing the responses of mothers and fathers. The overall results indicated the viability of constraining the factor loading to be the same across the groups. However, the CFI difference between Model 3 and Model 2 was higher than 0.001, which indicated that invariance of the intercepts was not complete across the two groups. The lack of invariance at this level does not permit the analysis of strict invariance, so we could not conduct analysis of differences in the latent or observed variables.

In the analysis of concurrent validity, moderate negative correlations were found for the EPDS (in mothers, $r = -0.41$, $p < 0.001$; and fathers, $r = -0.38$, $p < 0.001$), and positive correlations for the DAS (in mothers, $r = 0.39$, $p < 0.001$; and in fathers, $r = 0.44$, $p < 0.001$). Thus, higher scores in postnatal bonding were associated with lower scores in depressive symptomatology and higher dyadic adjustment.

# DISCUSSION

The objective of this study was to examine the psychometric properties of the Spanish version of the MPAS and the PPAS through the analysis of internal consistency, construct and concurrent validity, and thereby to contribute to clarifying the dimensions of the construct of postnatal bonding in mothers and fathers. The results of the study were obtained from a 15-item brief version, common for mothers and fathers, called PBS, which showed a good fit to the original three-factor structure proposed by *Condon & Corkindale (1998)*: seven items belonged to the quality of bonding dimension, four items to the absence of hostility dimension, and four items to the pleasure in interaction dimension. Worthy of note, to maintain the coherence of the study's theoretical framework and avoid further terminological ambiguity, the term "attachment" (in the original scale) was changed for "bonding." This latter term corresponds to the construct evaluated by the instrument.

The results obtained in the sample used in this study did not confirm the original three-factor solution of the MPAS (*Condon & Corkindale, 1998*) or the PPAS (*Condon, Corkindale & Boyce, 2008*), or the alternative models of the Italian version of the MPAS (*Scopesi et al., 2004*), or the Portuguese version of the two scales (*Nunes et al., 2014*; *Pires et al., 2014*). Therefore, none of the evidence found in previous research on the structure of the scale was observed in this study.

These results could indicate problems in the initial structure of the scales, due to the construction process: unstructured interviews with small samples (10 women in the case of the MPAS, and 15 men in the case of the PPAS), the reduction from 31 to 19 items in both scales due to statistical decisions (*Condon & Corkindale, 1998*; *Condon, Corkindale & Boyce, 2008*), and the heterogeneity in the response option alternatives.

Due to the lack of adequacy of previous confirmatory analyses, we considered the option of using a single version for men and women, using the 15 items common to both scales. This decision was initially proposed for theoretical reasons, and we subsequently confirmed its psychometric properties.

The theoretical reasons that justified this proposal are as follow. First, most psychological constructs are evaluated using instruments common to men and women, despite recognized gender differences (e.g., emotional intelligence; *Gartzia et al., 2012*). In this sense, it is understood that the essence of the concept is the same, although there may be differences in its expression. Therefore, in this way, the postnatal bonding construct could be assessed using a common instrument for men and women. Secondly, to separate an instrument by sex variable—men and women—is made difficult due to the gender identity variable, since this variable may be more important than sex to explain the differences between men and women (*Gartzia et al., 2012*). Thus, having a postnatal bonding's instrument which is suitable both for men and women, avoids the possibility of mistaking due to disregarding the distinction between sex and gender identity. Thirdly, disposing of common tools made up of the same items permits direct comparison between the bonding of men and women. In this sense, *De Cock et al. (2016)* defended the need for an instrument that evaluates prenatal and postnatal bonding conjointly in order to advance in parental bonding research. Likewise, this same argument can be used for postnatal bonding in men and women. Fourthly, different studies have used the same instrument of postnatal bonding in men and women (*Edhborg et al., 2005*; *Hall et al., 2015*; *Salian & Shah, 2017*), despite the fact that no studies have been found that specify the adaptation of this instrument to fathers.

In addition, it was considered necessary to use a panel of experts to obtain more evidence on the degree of relevance of the items both in men and women. The experts fully agreed that there were no differences in the degree of relevance of the items for men and women.

Therefore, evaluating these arguments, we considered that, at a theoretical level, a common instrument for men and women was adequate, and this was supported empirically.

We confirmed the three-factor structure of the original MPAS, the first instrument created by the authors (*Condon & Corkindale, 1998*), showing adequate fit index values. The majority of the standardized coefficients were statistically significant, with the exception of four items (items 4, 9, and 19 in the case of mothers, and Item 4 in the case of fathers). It will be necessary to analyze the functioning of these items in future studies.

The results of the global Cronbach alpha coefficients in the total sample were adequate (0.70 for mothers and 0.78 for fathers) and similar to the values found in the original versions (*Condon & Corkindale, 1998*; *Condon, Corkindale & Boyce, 2008*) and in the Italian version (*Scopesi et al., 2004*). However, the Cronbach alpha for the subscales showed lower values, above all in the case of mothers. In this case, we could not compare it with the original version or the Italian version, because these findings were not indicated. On the other hand, the Portuguese version (*Nunes et al., 2014*) and the Belgian version (*Van Bussel, Spitz & Demyttenaere, 2010*) of the MPAS also reported values similar to those obtained in the internal consistency of the subscales.

Due to the limitations of this method for reliability analysis (*Domínguez-Lara & Merino-Soto, 2015*; *Lozano, García-Cueto & Muñiz, 2008*), the CR was used. It is based on a structural equation modeling approach and represents a more accurate alternative for calculating reliability (*Peterson & Kim, 2013*). In this sense, as *Raykov (2001)* comments, the modeling of structural equations has the ability to empirically assess and overcome some of the limitations of the alpha coefficient. The CR results were optimal in the subscales quality of bonding (mothers, 0.74; fathers, 0.80), absence of hostility (mothers, 0.93; fathers, 0.94) and pleasure in interaction (mothers, 0.83; fathers, 0.90). Therefore, the CR results indicate that the reliability of the different subscales is adequate. We considered it necessary to maintain both procedures, coefficient alpha and CR, to allow comparing the results with previous studies and provide all necessary data for future research regarding the scale's reliability.

With regard to the scale invariance analyses carried out, we confirmed configural variance, which indicates that the construct was conceptualized in the same way in both samples (*Cheung & Rensvold, 2002*); that is, we concluded that mothers and fathers shared the same definition of bonding. We also observed metric invariance, which indicates that the participating mothers probably interpret each item that makes up the scale in the same way as the fathers do. However, it was not possible to confirm complete or partial scalar invariance between these two groups. The lack of invariance in this model may be due to the difference in the sample size of each group, as the size of the group of mothers was twice as large as that of the fathers. As a result, the strict invariance hypothesis is not supported.

Concurrent validity, measured through postnatal depression and dyadic adjustment, constructs used in the original PPAS (*Condon, Corkindale & Boyce, 2008*), showed high associations in the same line of the scientific literature and were consistent with the hypotheses proposed in the study. These results support the validity of the scale.

### Limitations and suggestions for future research

There were several limitations to this study. Firstly, the study data were obtained through self-administered questionnaires, which can lead to response bias such as social desirability. To reduce this bias, the combined use of self-reported questionnaires was suggested along with other types of measures to evaluate the postnatal bond, such as clinical interviews (e.g., The Stafford Interview, *Brockington et al., 2017*; YIPTA, *Leckman et al., 1994*) and direct observations (e.g., CIB, *Feldman, 1998*, BMIS, *Kumar & Hipwell, 1996*). In this sense, we consider the need to carry out more in-depth research on the relationship between social desirability and postnatal bonding, as well as related variables. Secondly, the participation of fathers and mothers was unequal and the response rate was low. There are usual limitations associated with studies in this field, but they remain relevant. Thirdly, the homogeneity of the sample characteristics did not allow generalizing the results. It would be interesting to use more heterogeneous samples in terms of age, educational level, income, nationality, family structure, context, and associated problems. Additionally, it would be desirable to test the instrument's performance at different stages of the child's life or following interventions, so as to assess the instrument's sensitivity to change.

We recommend two future lines of research. On the one hand, a theoretical reflection on the concept of postnatal bonding due to: (a) epistemological inconsistencies found in the review of scientific literature; (b) inconsistencies related to the factor structure of instruments assessing this concept; and (c) the need for further understanding the relationship between gender differences and postnatal bonding, as well as the analysis of the construct considering sociodemographic and cultural variables. On the other hand, as a result of the above, we recommend continuing to study the application of the new scale, including the removed items that belonged exclusively to the mother's or father's scale, or adding new items deriving from further theoretical reflection. In this way, a more relevant and complete construct can be obtained thus contributing to increasing the reliability of the scale.

## CONCLUSIONS

The new version of the scale of postnatal bonding, showing appropriate structural and concurrent validity and adequate reliability, is suitable for use in mothers and fathers. It evaluates the postnatal bond and allows comparing that of mothers and fathers: this contributes to progressing in perinatal mental health and in understanding gender differences in postnatal bonding. In this regard, we emphasize the need to include fathers in assessment and intervention programs within the context of transition to parenthood, thus favoring a systemic perspective on the family.

## ACKNOWLEDGEMENTS

The authors thank Alejandro Cerezo Munuera for his collaboration in the translating process of the scale, to the institution and all the professionals who participated, as well as to all the couples who contributed to the study.

### Funding

This project was supported by the General Sub-Directorate for Evaluation and Promotion of Research (Institute of Health Carlos III, ISCIII) and co-funded by the European Regional Development Fund (FEDER) (No. PI14/01549). The funders had no role in study design, data collection and analysis, decision to publish, or preparation of the manuscript.

### Grant Disclosures

The following grant information was disclosed by the authors:
The General Sub-Directorate for Evaluation and Promotion of Research (Institute of Health Carlos III, ISCIII).
The European Regional Development Fund (FEDER): PI14/01549.

### Competing Interests

The authors declare that they have no competing interests.

### Author Contributions

- Anna Riera-Martín conceived and designed the experiments, performed the experiments, analyzed the data, contributed reagents/materials/analysis tools, prepared figures and/or tables, authored or reviewed drafts of the paper, approved the final draft.
- Antonio Oliver-Roig conceived and designed the experiments, performed the experiments, analyzed the data, contributed reagents/materials/analysis tools, prepared figures and/or tables, authored or reviewed drafts of the paper, approved the final draft.
- Ana Martínez-Pampliega conceived and designed the experiments, performed the experiments, analyzed the data, contributed reagents/materials/analysis tools, prepared figures and/or tables, authored or reviewed drafts of the paper, approved the final draft.
- Susana Cormenzana-Redondo conceived and designed the experiments, performed the experiments, analyzed the data, contributed reagents/materials/analysis tools, prepared figures and/or tables, authored or reviewed drafts of the paper, approved the final draft.
- Violeta Clement-Carbonell analyzed the data, authored or reviewed drafts of the paper, approved the final draft.
- Miguel Richart-Martínez conceived and designed the experiments, performed the experiments, analyzed the data, contributed reagents/materials/analysis tools, prepared figures and/or tables, authored or reviewed drafts of the paper, approved the final draft.

### Human Ethics

The following information was supplied relating to ethical approvals (i.e., approving body and any reference numbers):

The study was approved by the Ethical Committee of Clinical Research of the General Direction of Public Health and Higher Center of Research in Public Health (CEIC-DSGSP/CSISP), attached to the Health Council of the Valencian Community.

## Data Availability

The raw data are provided in the Supplementary Files.

## Supplemental Information

Supplemental information for this article can be found online at http://dx.doi.org/10.7717/peerj.5980#supplemental-information.

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
