# Peer review of "A single Spanish version of maternal and paternal postnatal attachment scales: validation and conceptual analysis"

_PeerJ, doi:10.7717/peerj.5980_

## Round 0.1 · original submission · Major Revisions

Dear authors,

Your paper has been reviewed by 3 experts in the topic and they have found scientific merit in your work. However, they have suggested some major changes which you must apply in a revised version of your manuscript.

With respect and warm regards,
Dr Palazón-Bru (academic editor for PeerJ)

·

Basic reporting

The Authors present a Spanish validation of a new version of the Maternal and Paternal Postnatal Attachment Scales, usable both with mothers and fathers.
The manuscript covers a relevant topic within the aim and the scope of PeerJ-Section Brain and Cognition – Psychology.
The language, specifically the grammar, is of good quality. Nevertheless, some inaccuracies should be corrected:
line 26 I think that “evidence” is more appropriate than “evidences”
line 40 “a” should be “A”
line 57 no full stop after “introduction”
line 84-line 86-line 87 “problems” should be replaced by more accurate words
line 127 I think that comma is preferable to “and” before “affection”
line 152 no full stop after “Materials & Methods”
line 155 “were” should be “was”
line 164 I think that “cognitive” is not the appropriate term
line 266 no comma after “Sierra”
line 269 colons are preferable to comma
line 280 no full stop after “results”
line 323 I think that Figure is preferable to “figure”
line 353 no full stop after “Discussion”
line 374 I think that “its” is preferable to “their”
line 442 no full stop after “Conclusions”
Table 1 – note- “Standard Deviation”, not “Standard Desviation”
The structure of the paper is correct, figures and tables are accurate, and raw data are available.
The topic is well addressed and the authors’ quotations are relevant. The Authors provide a sufficient literature review about the construct of parental bonding, while they could provide more details about the findings from previous studies using MPAS and PPAS both in nonclinical and in clinical groups.

Experimental design

The manuscript covers a relevant topic within the aim and the scope of PeerJ-Section Brain and Cognition – Psychology.
The research question is well defined and relevant, the authors clearly state how their research fills an identified knowledge gap. The methods and the procedure are accurate and well described, statistical methods are valid.
Line 48 and line 348 - r=.41 and r=.30 are interpreted as moderate-high correlations; according to Guilford (1956) they are moderate correlations, not high.
The work has been conducted in conformity with the ethical standards of the field. Identifiable info has been removed from all files.

Validity of the findings

Data is statistically sound and controlled.
The Authors discuss adequately the research questions and conclusions are strongly supported by data. In the limitations section the authors could emphasize how much a self-report about parental bonding may be susceptible to social desirability and how much future studies are needed in both clinical and non-clinical samples. Moreover, the authors do not take into account that a parent's awareness of his normal, albeit low and controlled, feelings of hostility towards the child may actually be an indicator of a healthy parental bond while the failure to recognize normal feelings of hostility towards the child or high pleasure self-reported in interaction with the child can hide anger or hostility that can not be recognized due to intolerable feelings of guilt and shame.
Lastly, the Authors might add the self-report as an appendix.

·

Basic reporting

Some sections of the paper should be reviewed by a native english speaking colleague (ex. in the abstract, lines 25 -27 allude to the negative impact of postnatal bond in parents mental health and development of the child). Being the first lines and more important ones, it should be written in a more clear and unambiguous way.
The authors present good background based on relevant references in this area of knowledge, methodological option are supported. The work is pertinent, contributing to measurement validation to Spanish of an instrument used in attachment studies worldwide. It also points out the urge to discuss important theoretical issues, namely the distinction between bonding and attachment.
The structure follows standards and APA style. Tables and figures are well presented. However figure 1 has no caption.
Methods chosen are adequate (inter-rated consensus and confirmatory factor analysis - CFA). Authors should consider reviewing their reliability analysis. Cronbach’ Alpha values for both MPAS and PPAS sub-scales.

Experimental design

1. The current work complies to ethical standards, and an adequate analysis was carried out. Authors present sufficient information about methods, procedures and data analysis used to replicate the study.
Fathers and mothers groups are not equal numbered.
Recommendation: Although not essential for main purpose of the study, it should be added to the studies´ limitations. Authors present a satisfactory sample size. Nevertheless, dropouts rate should be indicated (line 213) and discussed further in the limitations section. The inclusion of 25 participants with other nationalities should also be justified (table 1).

2. Using the Delphi inter-rater method was a good option to increase validity of the back translation process. MPAS e PPAS global scale Cronbach’s alpha values are good. Although sub-scales values are very low andCronbach’s alpha is considered by some authors (see McDonald,1999; Maroco, 2006) to be conservator when applied to multi factor structure measures. inferior to usual standards (<.6). Cronbach’s alpha is considered by some authors (see McDonald,1999; Maroco, 2006) to be conservator when applied to multi factor structure measures.
Recommendation: Variance analysis could help to solve this out, since it´s more appropriate to study multiple factor reliability. In alternative, a single factor structure should be use, as suggested in the conclusion section. In a previous Portuguese study, 6 common items to MPASS e PPAS were used in CFA, including only the ones with higher eigenvalues to conducted a structural equation model (consider Brites, Pires, Nunes, Hipólito & Vasconcelos, 2016).

3. In both PPAS and MPAS, item 03 presents high kurtosis and skewness values (tables 2 and 3). Authors should clarified item 3 inclusion and the inclusion of items with lower factor loadings (e.g.: MPAS 3, 4,14, 19, 9, 2; PPAS 4 in figures 2 and 3).

Validity of the findings

Findings are robust and useful for the discussion on parents’ bonding vs attachment to the baby concept. It also contributes to the discussion regarding psychometric properties of Condon’s attachment measures MPAS e PPAS in a multicultural perspective, and it´s vulnerabilities. Since the final Spanish version uses 15 unaltered items of the original scales, it becomes difficult to figure out why the name was modified to “Postnatal Bonding Scale” (line 223). Changing only the name does not bring any added value per itself to the theoretical discussion of attachment vs bonding.
Recommendations: Discussion should emphasize more the second objective of the study (line 151): how does theses findings contribute to the concept discussion relating to previous studies presented in the background section. Authors should mantain original scale name (the title of the article is “A single Spanish version of Maternal and Paternal 1 Postnatal Attachment Scales: validation and conceptual analysis”).

References

Brites R., Pires M., Nunes O., Hipólito J., Vasconcelos M.L. (2016). Health Care Climate, Posttraumatic Stress Disorder and Mothers and Fathers’ Attachment to their Babies . In EADP (Ed.). Proceedings 17th European Conference on Development Psychology (pp.179-184). Bologna, IT: MEDIMOND s.r.l. ISBN 978-88-7587-733- 0
McDonald, R. P. (1999). Test theory: A unified treatment. Mahwah, NJ, US: Lawrence Erlbaum Associates Publishers.
Marôco, J. & Garcia-Marques (2006). Qual a fiabilidade do alpha de Cronbach? Questões antigas e soluções modernas?. Laboratório de psicologia, 4(1), 65-90.

Additional comments

The article is well written and the research performed is incremental. Nevertheless some parts of the paper should be reviewed by an English native colleague. Moreover, the content of some sections should be better justified. In order to support the quality improvement of this paper, I've included some recommendations of changes.

·

Basic reporting

The manuscript is, as far as I can judge, written in a clear English language. Regarding the introduction, it gives a good picture of the literature and research, framing the study. The paper appears to conform to the standards of the journal, the authors use Conclusion instead of Discussion in the Abstract, it is however not required according to the guidelines. I do consider the figures relevant for the article, adequately described and labeled. Tables 2 and 3 have less margins than recommended. All data sets used in the study are available.

Experimental design

The research presented in the paper is within the scope of the journal. Research questions implicit described on lines 149-151, relevant and meaningful. The study is performed according with technical and ethical standard. Methods well described admitting replication.

Validity of the findings

The impact and novelty with this study, i.e. a common measure for both mothers and fathers, is highlighted in the Discussion and Conclusions. Whetehr the data presented is robust and statistically sound depends on considerations regarding respones rate and that the data is not normally distributed.

Additional comments

I have some comments or questions that may be rather specific than general. What was the rationale for applying confirmatory factor analyses instead of exploratory factor analyses?
The authors do not mention anything about the participation rate that is presented in Figure 1. The response rate on the questionnaire seems to be quite low, 38.5% for women and 28.5% for men. Could this have a bearing on the results? Do the non-respondents differ from the respondents regarding socio-demographic characteristics? It appears as possible to look at this according to the supplied data files.
Characteristics of the sample are included in Table 1 and my question is if they could be related to the scales? Although it appears on the response pattern as the sample is quite homogenous, could it have some variation in relation to sociodemographic characteristics?
Could the sentence on the lines 236-237 be elaborated, how was the results consistent with the literature?
Concerning the lines 282-287 and Tables 2-3, the authors mention a kurtosis index of 2, what is this index based on? Any reference? How about an index regarding skewness? Is it possible to clarify what the values 1-5 in Tables 2-3 means? It is also unclear what DT stands for, could it be SD? Likewise would it be helpful with a clarification of Cu, is it kurtosis? Did the authors consider to exclude any items based on the descriptive analysis? If the values for skewness and kurtosis was critical as it appears to be for MPAS03 for example.
On line 334 does not the values for standardized coefficients for items 9 and 19 correspond to the values presented in Figure 2.
Is it possible to develop or clarify the theoretical reasons for a common measure for bonding on the lines 376-386? I have difficulties to follow the logic in the first two reasons in relation to a common measure.
Regarding limitations the authors mention self-administered questionnaires, is it possible to measure bonding in other ways that could be suggested for future research? Should the low participation rate be discussed as a limitation?

---

## Round 0.2 · accepted · Accept

Dear authors,

I am pleased to inform you that your paper has been accepted for publication in PeerJ.

Congratulations!

With respect and warm regards,
Dr Palazón-Bru (academic editor for PeerJ)

# ·

Basic reporting

The Authors present the revised version of a study about a Spanish validation of a new version of the Maternal and Paternal Postnatal Attachment Scales, usable both with mothers and fathers.
The language is now of good quality, the authors corrected all the previous inaccuracies.
The structure of the paper is correct, figures and tables are accurate, and raw data are available.
The topic is well addressed and the authors’ quotations are relevant. According to the suggestion, the Authors improved the introduction, by citing more pertinent and relevant previous studies.

Experimental design

The manuscript covers a relevant topic within the aim and the scope of PeerJ-Section Brain and Cognition – Psychology.
The research question is well defined and relevant, the authors clearly state how their research fills an identified knowledge gap. The methods and the procedure are accurate and well described, statistical methods are valid. The Authors corrected the inaccuracies found in the earlier version.

Validity of the findings

Data is statistically sound and controlled.
The Authors discuss adequately the research questions and conclusions are strongly supported by data.
They significantly improved the limitations section according to the recommendations.

·

Basic reporting

No comment

Experimental design

The authors has reinforced the reliability of the insturment.

Validity of the findings

I consider that the authors have corresponded to the comments given from the reviewers.